# Structural basis for SARS-CoV-2 envelope protein recognition of human cell junction protein PALS1

Jin Chai [1], Yuanheng Cai[2], Changxu Pang[1], Liguo Wang[3], Sean McSweeney[4], John Shanklin[1,2] & Qun Liu [1,4✉]

The COVID-19 pandemic, caused by the SARS-CoV-2 virus, has created global health and economic emergencies. SARS-CoV-2 viruses promote their own spread and virulence by hijacking human proteins, which occurs through viral protein recognition of human targets. To understand the structural basis for SARS-CoV-2 viral-host protein recognition, here we use cryo-electron microscopy (cryo-EM) to determine a complex structure of the human cell junction protein PALS1 and SARS-CoV-2 viral envelope (E) protein. Our reported structure shows that the E protein C-terminal DLLV motif recognizes a pocket formed exclusively by hydrophobic residues from the PDZ and SH3 domains of PALS1. Our structural analysis provides an explanation for the observation that the viral E protein recruits PALS1 from lung epithelial cell junctions. In addition, our structure provides novel targets for peptide- and small-molecule inhibitors that could block the PALS1-E interactions to reduce E-mediated virulence.

[1] Biology Department, Brookhaven National Laboratory, Upton, NY, USA. [2] Biochemistry and Cell Biology Department, Stony Brook University, Stony Brook, NY, USA. [3] Laboratory for Biomolecular Structure, Brookhaven National Laboratory, Upton, NY, USA. [4] NSLS-II, Brookhaven National Laboratory, Upton, NY, USA. ✉email: qunliu@bnl.gov

Severe acute respiratory syndrome coronavirus 2 (SARS-CoV-2) is a causative agent for the COVID-19 pandemic that is disrupting human health and global economy. Although 80% of COVID-19 patients display mild or no symptoms, 20% developed serious conditions mostly in the population of elderly person and those with underlying preexisting medical conditions. The virus has caused >3 million deaths and >160 million cases worldwide. Most deaths were associated with an acute respiratory distress syndrome (ARDS) and tissue damage linked to virus-induced hyperimmune responses[1].

SARS-CoV-2 and SARS-CoV-1 genomes encode a small envelope (E) protein that is a critically important component in assembly, release, and virulence phases of the viral life cycle[2]. SARS-CoV-2 E is composed of 75 amino acid residues with two distinct domains: an N-terminal transmembrane (TM) domain followed by a C-terminal domain. E is a multifunctional protein. Besides its structural roles required to induce membrane curvature for viral assembly in cooperation with the viral membrane (M) protein, E mediates host immune responses through two distinct mechanisms: a pore-forming TM domain related to the activation of NLRP3 inflammasome[3]; and a PDZ (PSD-95/Dlg/ZO-1)-binding function via its C-terminal domain[4,5]. Structurally, the TM domain of SARS-CoV-2 E forms a pentameric ion channel, similar to that of SARS-CoV-1 E[6,7]. However, the C-terminal domain has no well-defined structure, perhaps due to the lack of a stable complex.

In humans, there are ~150 unique proteins encoding one or more PDZ domains. These PDZ domains contain 80–110 amino acid residues, and are essential in regulating human immune responses and numerous physiological and pathological activities[8]. PDZ-domain-containing proteins in cell junctions have been hijacked by various viruses to potentiate their virulence[8]. Both SARS-CoV-2 and SARS-CoV-1 E proteins harbor a PDZ-binding motif (PBM) at their C-termini. Although the exact mechanism is unknown, interactions between the PBM and a human cell junction protein, PALS1, showed that E causes the relocation of PALS1 from the cell junction to the endoplasmic reticulum–Golgi intermediate compartment (ERGIC) site, where E is localized, and viral assembly and maturation occur[4].

PALS1 is an integral part of an apical cell polarity complex, consisting of Crumbs, PALS1, and PATJ[9]. Under physiological conditions, PALS1 interacts with the Crumbs C-terminus (Crb-CT) through the PSG module[10] and interacts with PATJ through its N-terminal L27 domain[11] (Supplementary Fig. 1a). In SARS-CoV-2-infected lung epithelial cells, the replication and transcription of the virus genome produces a high load of the E protein, which localizes to the ERGIC region for viral assembly and budding[12]. It is proposed that the specific interactions between E and PALS1 recruit PALS1 to the site of virus assembly, and could possibly disrupt the polarity complex and vascular structure[4]. Consequently, the inter-epithelial junctions loosen and leak. The leaking junctions may likely promote local viral spread, flow of fluid, and multiple types of immune cells (such as monocytes and neutrophils) into lung alveolar spaces (Supplementary Fig. 1b). In addition to PALS1, E also interacts to PDZ-containing adhesion junction protein syntenin[5], tight junction protein ZO-1[13], and other cell junction proteins[14]. The relocation of these cell junction proteins in lung epithelial cells might contribute to vascular leakage, diffuse alveolar damage, cytokine storm initiation, and ARDS, commonly leading to death in elderly COVID-19 patients and those with underlying conditions[2].

The PBM in E contains four conserved residues (DLLV), and is conserved between SARS-CoV-2 and SARS-CoV-1 viruses. The motif appears to play a critical role in virulence because mutants without the PBM are either attenuated or nonviable[5,15]. Binding assays, using C-terminal peptides of SARS-CoV-2 E and

SARS-CoV-1 E, show enhanced binding affinity of SARS-CoV-2 E peptide to the PDZ domain in PALS1[16]. However, there is a lack of structural information to define such protein–protein interactions, which hinders further understanding of the mechanisms of the E-mediated virulence. In this work, we describe the structure of the PALS1–E complex to define the mechanism of recognition of the PALS1 PDZ and SH3 domains by the C-terminal PBM of the E protein.

## Results

**Production of the PSG–Ec18 complex.** PALS1 contains five domains, two N-terminal L27 domains and three C-terminal domains, PDZ, SH3, and GK (named as PSG). To improve protein stability, we expressed and purified the PSG (residues 236–675) without a loop between the SH3 and GK domains[10] (Supplementary Fig. 2a). The expressed protein was purified by Ni-NTA (nickel-nitrilotriacetic acid) affinity resins followed by size-exclusion chromatography (SEC; Supplementary Fig. 2b). Based on the SEC analysis, we found that majority of PSG is a dimer (Supplementary Fig. 2c).

To study the structural basis for recognition of PALS1 PSG by the SARS-CoV-2 E C-terminal domain, we synthesized an E C-terminal 18-amino acid peptide (Ec18) containing the PBM (Supplementary Fig. 2a). To check the binding affinity between Ec18 and PSG, we labeled PSG using a fluorescence dye and titrated it using a serial dilution of Ec18. Using a microscale thermophoresis (MST) method[17], we determined the $K_d$ at 11.2 µM (Supplementary Fig. 2d). Our measured value is consistent with the binding affinity using a 10-aa peptide and the PDZ domain alone, where the reported $K_d$ is 40 µM[16]. Considering the low affinity between Ec18 and PSG, we used a high ratio of Ec18 for complex formation by incubating purified PSG with Ec18 at a molar ratio of 1:10 for 2 h at room temperature.

**Structure determination.** We subjected the PSG–Ec18 complex to analysis using single-particle cryo-EM. Our initial 2D class averages showed a preferred particle orientation. To get additional views of the complex, we performed detergent screening and found that the inclusion of 0.05% CHAPS allowed PSG–Ec18 particles to adopt random orientations (Fig. 1a), and helped us obtain multiple views after 2D class averaging (Fig. 1b, e). We optimized our particle-picking procedure using a local dynamic mask for defocus-based particle picking[18]. After iterative 2D and 3D classifications and refinements with per-particle CTF and Bayesian polishing (Supplementary Fig. 3) with Relion3 and CryoSPARC[19,20], we obtained a final reconstruction at 3.65 Å (Fig. 1c), using Fourier shell correlation of 0.143 as a cutoff (Fig. 1d). The map shows clear secondary structures and side chains that allowed us to build and refine atomic models (Supplementary Fig. 4).

**Structure of the PSG–Ec18 complex.** The solved structure contains a dimer of PSG and a single Ec18 (Fig. 2a). In one PSG monomer, the PDZ, SH3, and GK domains were observed; while in the other monomer, the PDZ domain was missing. However, in our initial 3D classification, we observed a class with a highly disordered region corresponding to the missing PDZ domain (Supplementary Fig. 3), suggesting that PSG–Ec18 interactions are transient and dynamic.

In our structure, Ec18 is inserted in a hydrophobic pocket between the PDZ and SH3 domains through the PBM (72DLLV75) (Fig. 2b, c). The density coverage for residues Leu74–Val75 on the Ec18 and Phe318 on the PDZ domain are well defined, and help position Ec18 in the binding pocket. Residues Phe318, Leu321, Leu267, Pro266, and Val284 from

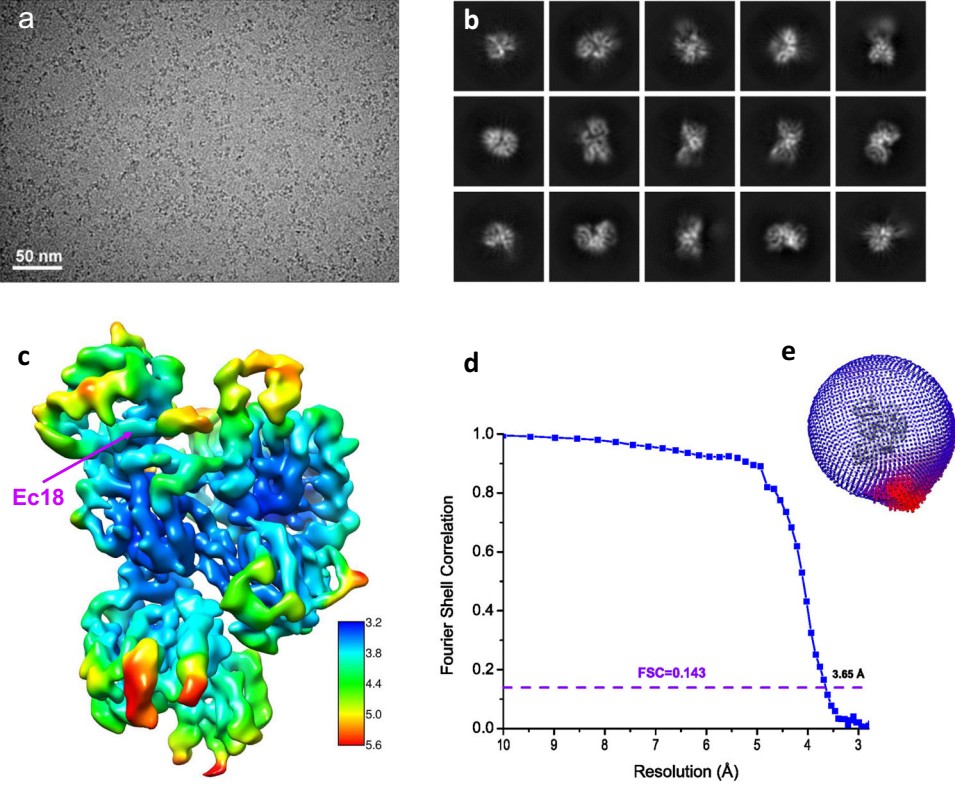

**Fig. 1 Structure determination by single-particle cryo-EM. a** A typical motion-corrected cryo-EM micrograph from a total of 12,826 micrographs. **b** 2D class averages. **c** Reconstructed map colored with local resolutions. **d** Fourier shell correlation (FSC) curve for the 3D reconstruction to determine the structure resolution. **e** Orientation distribution for particles used in 3D reconstruction.

PDZ, and Leu369 and Leu403 from SH3 are involved in forming the hydrophobic-binding pocket. Phe318, Leu369, and Leu403 in PALS1, and Leu74 and Val75 in Ec18 have side-chain densities, consistent with roles in the formation and recognition of the binding pocket, respectively. Among these residues, Phe318 is sandwiched by two hydrophobic residues Leu73 and Val75 in Ec18, representing another notable recognition feature.

There are two SH3 and two GK domains in the structure. The overall structure for the GK and SH3 domains are similar: the root mean square deviation (RMSD) is 1.18 Å for 205 $C_\alpha$ atoms in the GK domain and 1.23 Å for 65 $C_\alpha$ atoms in the SH3 domain. Nevertheless, we found conformational changes for two SH3 loops associated with Ec18 binding (Fig. 2d). One loop containing residue Leu403 moved as much as 4.5 Å; and the other loop containing Leu369 also moved so that Leu369 has a closer engagement with Leu74 from Ec18. The interactions between the Ec18 and SH3 domains are unexpected, and we propose that residues Leu369 and Leu403 from the SH3 domain further enhance Ec18 binding to the PDZ domain.

**Structural comparison with the PSG–Crb-CT complex.** The binding affinity between Ec18 and PALS1 is weaker than the interaction between PALS1 and its physiological ligand Crb-CT[10]. Nevertheless, viral E protein competes with Crb-CT for binding to and relocalization of PALS1[4]. Using single-particle cryo-EM, we captured the transient complex between Ec18 and PALS1 for structural analysis. To understand the structural basis of Ec18–PALS1 binding and its implication with respect to virulence, we compared this complex with the X-ray structure of the PSG–Crb-CT complex (PDB code 4WSI). As shown in Fig. 3, Crb-CT interaction involves three domains (PDZ, SH3, and GK) of PSG, while Ec18 appears to interact with the PDZ and SH3 domains only. Consequently, in the PSG–Ec18 complex, the PDZ

and SH3 domains are rotated ~38° relative to the GK domain, likely due to the disengagement of PDZ and SH3 domains from the GK domain (Fig. 3a).

Based on the alignment of PSG–Ec18 and PSG–Crb-CT structures (Fig. 3b), the terminal isoleucine in Crb-CT is inserted deeply in the PDZ pocket. So, a peptide inhibitor with a C-terminal isoleucine, leucine, or even phenylalanine might penetrate the pocket deeper with a higher affinity. In addition, an arginine in Crb-CT PBM may interact with Phe318 unfavorably; changing this residue to a hydrophobic residue, such as leucine or phenylalanine may enhance its hydrophobic interactions with Phe318. Interestingly, through evolution in hosts, SARS-CoV-2 variants have acquired mutations in the PBM (72DLLV75) for viral fitness and virulence. Notable PBM mutations are D72Y, D72H, L73F, V75L, and V75F[21,22]. We therefore propose that a hybrid peptide inhibitor containing Crb-CT and viral PBM mutations would weaken the PALS1–E interactions and suppress E-mediated virulence.

**Discussion**

Virus–host interactions have been proposed to potentiate viral fitness and virulence[23]. Many viruses have developed strategies to hijack human PDZ-domain-containing proteins to increase their virulence and evade immune responses[24,25]. Mutations in viruses, including SARS-CoV-2, that convey a selective advantage with respect to replication, assembly, release, and spread can accelerate the viral life cycle. In this work, we provide a complex structure to show interactions between the SARS-CoV-2 E protein and human PALS1. The structure of the PSG–Ec18 complex allows us to explain the mechanism of E-mediated PALS1 relocation and virulence, as illustrated in Supplementary Fig. 1. In the structure, the E C-terminal PBM binds to a pocket formed by the PDZ and SH3 domains (Fig. 2c). Interestingly, when the PBM was deleted in SARS-CoV variants, the PBM was recovered from passage in

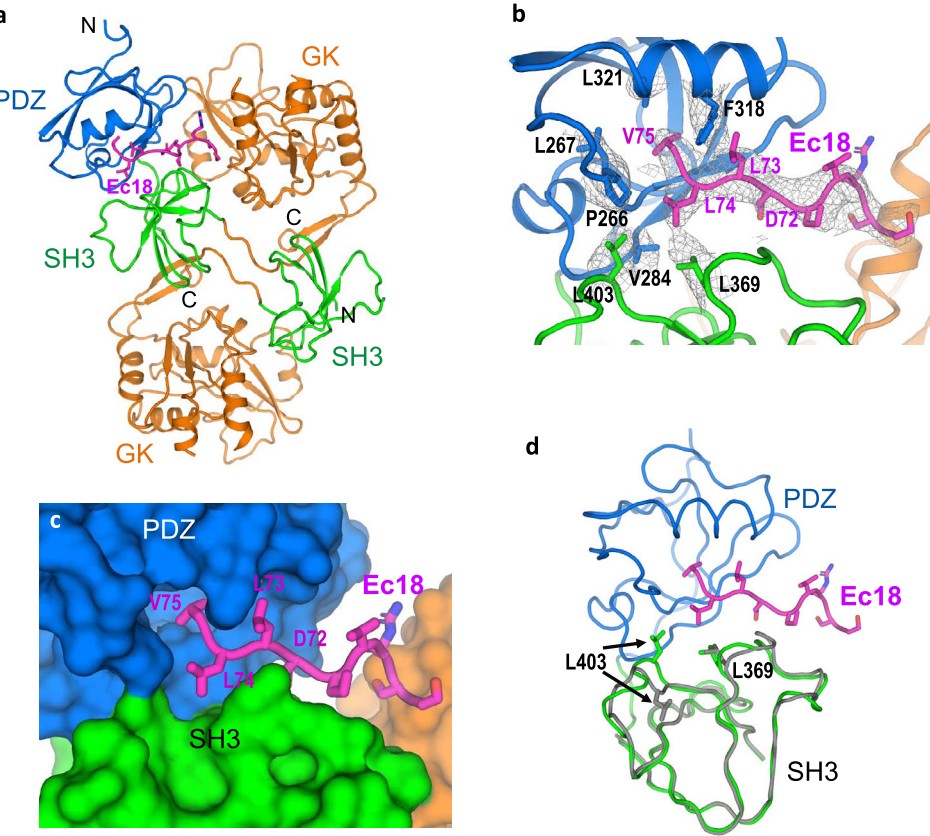

**Fig. 2 Recognition of the PALS1 PDZ and SH3 domains by the E PBM. a** Structure of the PSG–Ec18 complex shown as cartoons with different colors for different domains. Ec18 is shown as magenta sticks. **b** Binding site structure. Hydrophobic residues forming the binding pocket were shown as sticks. Potential density map for the binding site is shown as gray isomeshes contoured at 5.5σ. **c** Surface representation of the binding site with Ec18 showing as sticks. **d** Superimposition of the two PSG monomers to show conformational changes in SH3 domains. The PDZ domain in the second monomer is disordered.

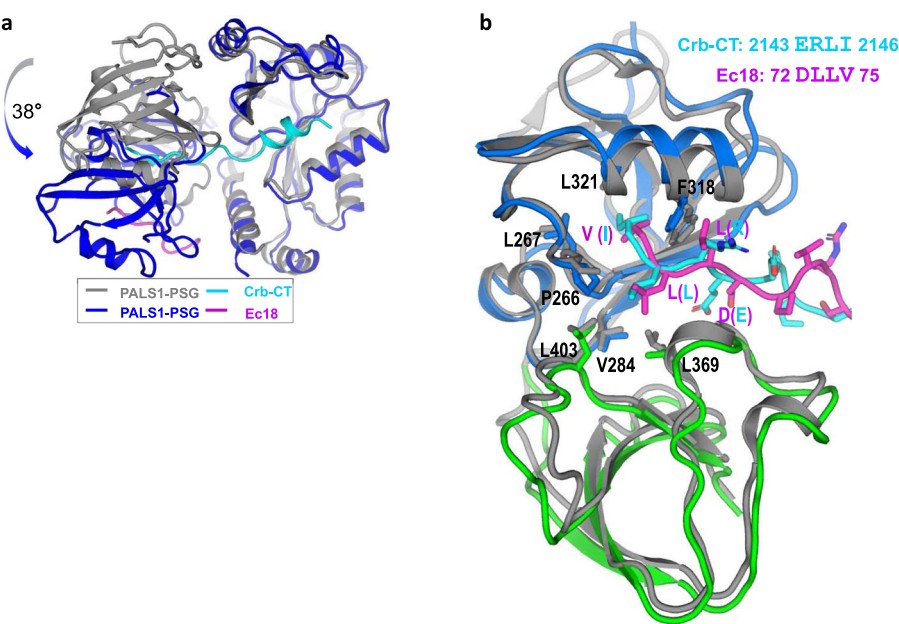

**Fig. 3 Structural comparison of PSG–Ec18 with PSG–Crb-CT. a** Alignment of the two complex structures based on the GK domain showing a relative rotation of ~38° for the SH3 and PDZ domains. **b** Structural superposition for the SH3 and PDZ domains showing the binding sites for Ec18 and Crb-CT.

the host, clearly demonstrating its involvement in viral fitness and virulence. In addition to E, SARS-CoV-2 and SARS-CoV-1 encode another PBM-containing protein ORF3a, which could both be involved in the recruitment of PDZ-containing proteins for viral fitness and virulence[26].

There are seven human coronaviruses (hCoVs), all expressing E proteins. However, only three (SARS-CoV-1, SARS-CoV-2, and MERS-CoV) have E-mediated virulence[2]. Multiple sequence alignment of the seven hCoV E proteins (Supplementary Fig. 5) shows that they all have a terminal hydrophobic residue (V, I, or F). Compared to their N-termini, sequence variations are larger at the C-termini including the PBM. The strongest virulent SARS-CoV-1 and SARS-CoV-2 have the four-residue PBM sequence DLLV. The DLLV motif does not present in MERS-CoV and four non-virulent hCoVs. Lower virulence of MERS-CoV shows conservation of two of the four residues of SARS-CoV-1 and SARS-CoV-2, i.e., DEWV, suggesting a role of a D at position 1 and V at position 4. For the four non-virulent strains, hCoV-OC43 and hCoV-NL63 share a V at position 4 and hCoV-HKU1 shares a D at position 1. The remaining non-virulent strain hCoV-229E shares no identical residues at position 1 or 4. In addition to the DLLV motif, previous reports show that SARS-CoV-1 mutant replacing DLLV by GGGG in E is attenuated, suggesting contribution of other C-terminal residues to virulence[27]. In our structure, we observed densities for 9 out of 18 residues from the Ec18 peptide (Fig. 2b). Residues immediately before the DLLV motif may module Ec18 flexibility, solubility, and affinity to PALS1. We hence propose the need of designing and developing longer peptides, in order to reduce E-mediated PDZ-binding and virulence.

## Methods

**Protein expression and purification**. The gene encoding PASL1–PSG domains (residues 236–675) with a deletion between 411–460 was codon optimized for bacterial expression, and synthesized and cloned into pET16-b by Genscript (www.genscript.com) with an N-terminal 10× his-tag followed by a tobacco etch virus (TEV) cleavage site.

The protein was overexpressed in *Escherichia coli* BL21 (DE3) pLysS at 16 °C for 18 h induced by addition of 0.4 mM IPTG (final) to the cell culture with an A600 of 1.0. Harvested cells were resuspended in extraction buffer containing 30 mM Tris, pH 7.5, 150 mM NaCl, 1.0 mM TCEP, and 0.2 mM PMSF. Cells were lysed using an EmulsiFlex-C3 Homogenizer (Avestin, Ottawa, Canada). After centrifugation at $26,000 \times g$ for 30 min, the supernatant was collected for affinity purification by Ni-NTA affinity chromatography (Superflow, Qiagen). The eluate was concentrated and buffer exchanged for tag removal by incubation with TEV protease overnight at 4 °C. The protein-containing solution was passed through Ni-NTA resin again to remove the cleaved tag and the protein flow-through fractions were collected, concentrated, and applied to a size-exclusion column (TSKgel G3000SW column, Tosoh Bioscience) pre-equilibrated with 25 mM Tris, pH 7.5, 100 mM NaCl, and 1 mM TCEP. Highly enriched protein was concentrated to ~10 mg/ml using an Amicon Ultra-15 centrifugal filter with a molecular cutoff of 30 kDa (Millipore, Inc).

**Cryo-EM sample preparation and data collection**. To make the PSG–Ec18 complex, we mixed PSG and synthesized Ec18 (Vivitide, Gardner, MA) at a molar ratio of 1:10 at a final concentration of 2 mg/ml. After incubation for 2 h at room temperature, we added 0.05% CHAPS (3-((3-cholamidopropyl) dimethylammonio)-1-propanesulfonate) to the sample immediately before applying 3 μl of the sample to a glow-discharged Quantifoil Au grid (0.6/1.0). Vitrification was performed using a ThermoFisher Mark IV vitrobot with a blotting condition of 3.5 s blot time, 0 blot force, and 100% humidity at 6 °C.

Cryo-EM data were collected with the use of a ThermoFisher Titan Krios (G3i) equipped with a Gatan K3 camera and a BioQuantum energy filter. With a physical pixel size of 0.684 Å, a total dose of 64 e⁻/Å² was fractioned to 52 frames under the super-resolution mode, using the ThermoFisher data acquisition program EPU. A total of 12,861 movies were collected with an energy filter width of 20 eV throughout the data acquisition. Data collection statistics are listed in Supplementary Table 1.

**Cryo-EM data analysis**. Beam-induced motion correction was performed using MotionCorr2[28] through a wrapper in Relion3[19], with a bin-factor of 2. Corrected and averaged micrographs were further corrected by CTF estimation using Gctf[29].

Micrographs with an estimated resolution worse than 4.5 Å were discarded from further processing. Particle picking was performed using Localpicker[18], which uses per-micrograph defocus values (estimated by Gctf) to set up picking parameters. We picked a total of 6,375,890 particles, extracted and binned them to 64 pixels with a pixel size of 2.736 Å.

We used CryoSPARC[20] and Relion3 for 2D and 3D class averages and 3D refinements. Specifically, we used 2D class averaging for initial particle cleanup, which resulted in 2,193,282 particles. Using these particles, we produced an initial 3D model in CryoSPARC and used the model to perform 3D classifications in Relion3 for five classes with a pixel size of 2.736 Å (Supplementary Fig. 3). Particles from the 3D class with the best structural feature as visualized in Chimera[30] were selected. A total of 715,010 particles were selected, re-centered, and re-extracted at 256 pixels with a pixel size of 0.684 Å.

Extracted particles were further auto-refined to convergence with Relion3 followed by a nonalignment 3D classification into three classes (Supplementary Fig. 3). Particles from the best class (7.2%) were selected for CTF refinement and Bayesian polishing in Relion3, and nonuniform refinement in CryoSPARC to reach a refined reconstruction at 3.65 Å resolution based on gold-standard Fourier shell correlation of 0.143 (Fig. 1d). Local resolutions were estimated using BlocRes[31]. Reconstruction statistics are listed in Supplementary Table 1.

**Model building and refinement**. To assist our model building and refinement, we sharpened the masked and filtered map using PHENIX[32] with a *B* factor of −100 Å². We used the PDB code 4WSI as a starting model, and built the model for PSG and Ec18 in COOT[33], and refined the model iteratively using PHENIX. The refined model[34] was validated using Molprobity[35], and the refinement statistics are listed in Supplementary Table 1.

**Microscale thermophoresis measurement**. The binding affinity between Ec18 and PALS1 PSG was measured using a Monolith NT.115 instrument (Nanotemper, Munich, Germany). Purified protein was buffer exchanged and covalently labeled using dye NT647, following manufacturer's protocol. The labeled protein was diluted 10× prior to measurement in assay buffer containing 50 mM Tris, pH 7.5, 100 mM NaCl, 1 mM DTT, 1 mM EDTA, and 0.05% Tween 20. Ec18 was dissolved in the assay buffer to a final concentration of 5.0 mM. Ten microliters of Ec18 was diluted 1:1 serially in the assay buffer and mixed with an equal volume of labeled PSG. The PSG–Ec18 samples were incubated for 10 min in dark at room temperature before MST measurements. For all MST measurements, we used a MST power medium, laser power 40%, and MST time 30 s. NanoTemper program MO. Affinity Analysis (Nanotemper, Munich, Germany) was used for data analysis and curve fitting with a $K_d$ model.

**Reporting summary**. Further information on research design is available in the Nature Research Reporting Summary linked to this article.

## Data availability
The three-dimensional cryo-EM density map has been deposited in the Electron Microscopy Data Bank under the accession number EMD-23665. Atomic coordinates have been deposited in the Protein Data Bank under the accession number 7M4R. Source data are provided with this paper.

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

## Acknowledgements

We thank Gongrui Guo for initial discussions on the project, LBMS staff for the help with the cryo-EM operation and data acquisition, and Computational Science Initiative (CSI) for support on computation. This work was supported in part by Brookhaven National Laboratory COVID-19 LDRD. LBMS is supported by the U.S. Department of Energy, Office of Science, Office of Biological and Environmental Research. J.S. was supported by Division of Chemical Sciences, Geosciences, and Biosciences, Office of Basic Energy Sciences, U.S. Department of Energy Grant DOE KC0304000.

## Author contributions

Q.L. designed the study and experiments. J.C., Y.C., C.P., L.W., and Q.L. performed the experiments. J.C., S.M., J.S., and Q.L. analyzed the data. Q.L. wrote the manuscript with help from the other coauthors.

## Competing interests

The authors declare no competing interests.
