## [Peer Review File · Nature Communications]

REVIEWER COMMENTS

Reviewer #2 (Remarks to the Author):

This manuscript reports that the coronavirus E protein C-terminal motif - DLLV - recognizes a pocket formed by hydrophobic residues from the PDZ and SH3 domains in PALS1. The authors use structural analysis to support their theory that the viral E protein recruits PALS1 from lung epithelial cells. They then hypothesize that this recruitment in lung epithelial cell junctions results in vascular leakage, lung damage, viral spread, and virulence. They also propose that the structural evidence provides novel targets for peptide- and small-molecule inhibitors that could block the PALS1-E interactions to reduce the E-mediated damage to vascular structures.

This article provides a novel finding that could influence the thinking in the field; could possibly change/add to the way we look at mitigating the damage caused by the pathogenic coronaviruses. The methodology is sufficiently explained for other to repeat.

This is an interesting article that sets out to explain the possible mechanisms linked to the coronavirus E protein viroporin and possible vascular leakage roles.

The authors should explain/speculate why, considering that the E protein is found in all human coronaviruses, we see this virulence and pathogenicity roles for SARS-CoV, SARS-CoV-2 (and possibly MERS) and not in the other 4 hCoVs; e.g. is the DLLV motif only present in SARS-CoV and SARS-CoV-2?

Reviewer #3 (Remarks to the Author):

This manuscript by Chai et al. reports the cryo-EM structure of the human cell junction protein PALS1 bound to an eighteen amino acid peptide from the SARS-CoV-2 envelope (E) protein. The study deals with a timely topic, but its significance and the new insight it provides is somewhat limited.

The authors claim "The structure provides an explanation for the observation that viral E protein recruits PALS1 from lung epithelial cell junctions resulting in vascular leakage, lung damage, viral spread and virulence". This claim is overstated. At best the manuscript provides a structural basis for the interaction between this fragment of the E protein and PALS1. The effect of this fragment through its binding to PALS1 was already known, and the manuscript, lacking new functional data, does not provide additional insight in this regard. This fact limits the impact of the manuscript.

Another claim is that the structure provides a framework for the design of peptide mimics and small molecule inhibitors that would prevent the interaction between PALS1 and the E protein. Here the issue is that the interaction with the 18 amino acid peptide of the E protein is low affinity, and the structure does not provide enough additional key insight to drive such efforts. In particular, given the fact that the structure of PALS1 bound to a fragment of its physiological ligand is already known, and thus the binding pocket has been previously described.

From a cryo-EM perspective it is commendable that the authors have been able to overcome issues of limited angular space coverage, low occupancy, and conformational heterogeneity to solve a 100 kDa protein to a respectable resolution. This being said, as mentioned above the x-ray structure of the PSG dimer bound to a peptide fragment from its physiological ligand was already known, and thus from a structural biology perspective the novelty is also limited.

The model presented in Fig. 3 is not founded in any specific new insight provided by the manuscript, and merely recapitulates what was already known about this system.

Overall the manuscript presents a study at an early stage, somewhat incipient, and lacking functional data. As a whole it has not been developed to a level one expects in order to warrant

publication in a top journal.

Referees' comments:

Reviewer #2 (Remarks to the Author):

This manuscript reports that the coronavirus E protein C-terminal motif - DLLV - recognizes a pocket formed by hydrophobic residues from the PDZ and SH3 domains in PALS1. The authors use structural analysis to support their theory that the viral E protein recruits PALS1 from lung epithelial cells. They then hypothesize that this recruitment in lung epithelial cell junctions results in vascular leakage, lung damage, viral spread, and virulence. They also propose that the structural evidence provides novel targets for peptide- and small-molecule inhibitors that could block the PALS1-E interactions to reduce the E-mediated damage to vascular structures.

This article provides a novel finding that could influence the thinking in the field; could possibly change/add to the way we look at mitigating the damage caused by the pathogenic coronaviruses. The methodology is sufficiently explained for other to repeat.

This is an interesting article that sets out to explain the possible mechanisms linked to the coronavirus E protein viroporin and possible vascular leakage roles.

Response: We thank the referee for positive comments on novelty and technical quality of our work.

The authors should explain/speculate why, considering that the E protein is found in all human coronaviruses, we see this virulence and pathogenicity roles for SARS-CoV, SARS-CoV-2 (and possibly MERS) and not in the other 4 hCoVs; e.g. is the DLLV motif only present in SARS-CoV and SARS-CoV-2?

Response: We thank the referee for raising this issue and as suggested we performed a comparison of the c-terminal PBM sequences of SARS-CoV, SARS-CoV-2, MERS-CoV and four other hCoVs (hCoV-229E, hCoV-NL63, hCoV-OC43 and hCoV-HKU1) as shown in **Supplementary Fig 5**. All seven hCoVs have a hydrophobic terminal residue (V, I, or F). The strongest virulence of SARS-CoV-1 and SARS-CoV-2 shows they have the same four-residue amino-acid sequence DLLV. The DLLV motif does not present in MERS-CoV and four non-virulent hCoVs. Lower virulence of MERS-CoV shows conservation of two of the four residues of SARS-CoV-1 and SARS-CoV-2, i.e., DEWV, suggesting a role of a D at position 1 and V at position 4. For the four non-virulent strains, hCoV-OC43 and hCoV-NL63 share a V at position 4 and hCoV-HKU1 shares a D at position 1. The remaining non-virulent strain hCoV-229E shares no identical residues at position 1 or 4. This new sequence analysis is shown in a new **Supplementary Fig. 5** and is discussed on **p.6** of the revised manuscript.

Reviewer #3 (Remarks to the Author):

This manuscript by Chai et al. reports the cryo-EM structure of the human cell junction protein PALS1 bound to an eighteen amino acid peptide from the SARS-CoV-2 envelope (E) protein. The study deals with a timely topic, but its significance and the new insight it provides is somewhat limited.

Response: Though vaccination may alleviate disease severity in COVID-19 patients, SARS-CoV-2 variants and their potential evasion of the immune system create an urgent imperative to develop alternative antiviral strategies. It is our view that the timely publication of this cryo-EM structure for the PALS1-PSG-Ec18 complex will help the community in better understanding the mysterious and missing structural link between the viral E protein and its virulence. It is likely that our structural work will inspire additional mechanistic studies and the associated development of antiviral drugs designed to disrupt viral/host interactions, in

particular interactions involving human lung epithelial cell junction proteins relating to COVID-19 disease severity.

The authors claim "The structure provides an explanation for the observation that viral E protein recruits PALS1 from lung epithelial cell junctions resulting in vascular leakage, lung damage, viral spread and virulence". This claim is overstated. At best the manuscript provides a structural basis for the interaction between this fragment of the E protein and PALS1. The effect of this fragment through its binding to PALS1 was already known, and the manuscript, lacking new functional data, does not provide additional insight in this regard. This fact limits the impact of the manuscript.

Response: As pointed out by the referee, our work provides a structural basis for the interactions between the C-terminal fragment of E with human PALS1. For the SARS-CoV-1 E protein, it has been shown that human cell junction protein PALS1 is recruited to the site where E is localized, and viruses replicate and assemble. SARS-CoV-2 E protein has an identical C-terminal DLLV motif as SARS-CoV-1 E protein. Therefore, we infer that SARS-CoV-2 E protein could also recruit PALS1 from epithelial cell junctions. We agree with the Referee that the effect of the fragment through its binding to PALS1 was already known. Nevertheless, presenting such a formal hypothesis is likely to inspire additional functional work that will lay the foundation for additional antiviral strategies for SARS-CoV-2. We acknowledge Referee 3's opinion that as presented, the text was somewhat overstated, and we have carefully edited the text to tone down our claims. (p. 1, abstract; p.5, discussion; text colored in blue).

Another claim is that the structure provides a framework for the design of peptide mimics and small molecule inhibitors that would prevent the interaction between PALS1 and the E protein. Here the issue is that the interaction with the 18 amino acid peptide of the E protein is low affinity, and the structure does not provide enough additional key insight to drive such efforts. In particular, given the fact that the structure of PALS1 bound to a fragment of its physiological ligand is already known, and thus the binding pocket has been previously described.

Response: It is true that the interactions between the viral E protein and PDZ domains are transient with micromolar affinities. However, such interactions are crucial for the fitness of virus. SARS-CoV-1 E competes with Crb-CT in a concentration-dependent manner to recruit PALS1 and to disrupt tight junction formation. Therefore, weak interactions between PALS1 and E can be used to help the development of effective inhibitors to reduce E-PALS1 interactions and virulence. In addition, as compared to the PSG-Crb-CT complex, our structure adds the following additional insights: 1) PALS1-Ec18 interactions require both PDZ and SH3 domains which was not expected until we solved the structure. Also, PALS1-Ec18 interactions do not require the GK domain and thus explain why the interactions are weak and transient. 2) Structural comparison between PALS1-Ec18 and the PSG-Crb-CT structures provides a foundation for developing effective inhibitors to selectively compete with the E-PALS1 binding while leaving PALS1 at tight cell junctions. To highlight these structural features in comparison to the PSG-Crb-CT structure, we moved **Supplemental Fig. 4** into the main figure and present it as **Fig. 3** and we added a section "Structural comparison with the PSG-Crb-CT complex" to the manuscript text. (p. 5).

From a cryo-EM perspective it is commendable that the authors have been able to overcome issues of limited angular space coverage, low occupancy, and conformational heterogeneity to solve a 100 kDa protein to a respectable resolution. This being said, as mentioned above the x-ray structure of the PSG dimer bound to a peptide fragment from its physiological ligand was already known, and thus from a structural biology perspective the novelty is also limited.

Response: We thank Referee 3 for positive comments regarding the technical contributions of our work. From a structural biology perspective, this is the first complex structure to demonstrate how SARS-CoV-2 uses its E protein to hijack the human cell junction protein PALS1. The prior knowledge of E-PALS1 interactions focused on the PDZ domain alone. Our structure shows that both the PDZ and SH3 domains are involved in defining the binding pocket that requires conformational changes of two SH3 loops. In addition, as we discussed above, our structure and associated analysis provides new insights for inhibitor design targeted at reducing the E-PALS1 interactions and E-mediated virulence. Therefore, scientifically it has significant novelty with respect to the SARS-CoV-2 pandemic.

The model presented in Fig. 3 is not founded in any specific new insight provided by the manuscript, and merely recapitulates what was already known about this system.

Response: The purpose of the model shown in **Fig. 3** was to help readers understand the PALS1-E interactions in a broader context. Based on reviewer comments, we have deemphasized it by moving it from the main text to Supplementary information and renamed it as **Supplementary Fig. 1**. Also, in recognition that the information is largely from prior studies we moved the associated text to Introduction (**p. 2-3**).

Overall the manuscript presents a study at an early stage, somewhat incipient, and lacking functional data. As a whole it has not been developed to a level one expects in order to warrant publication in a top journal.

Response: The main advance of this work is to report the first complex structure of SARS-CoV-2 Ec18 and human PALS1-PSG to provide the community with structural details that define their interactions. The structure helps to explain the E PBM-mediated virulence and provide atomic coordinates to assist computational simulations and modelling for the design and development of novel antiviral peptide and small-molecule inhibitors. We have revised our manuscript thoroughly to highlight the structural details of the SARS-CoV-2 Ec18 and human PALS1-PSG complex relative to that of the existing PSG-Crb-CT structure and toned down the previously overstated claims. As suggested by Referee 2, we expanded our sequence analysis to include other human coronavirus E proteins to provide new insights into virulence.